# Fermentation Conditions Affect the Synthesis of Volatile Compounds, Dextran, and Organic Acids by *Weissella confusa* A16 in Faba Bean Protein Concentrate

**DOI:** 10.3390/foods11223579

**Published:** 2022-11-10

**Authors:** Fabio Tuccillo, Yaqin Wang, Minnamari Edelmann, Anna-Maija Lampi, Rossana Coda, Kati Katina

**Affiliations:** 1Department of Food and Nutrition Sciences, University of Helsinki, FI-00014 Helsinki, Finland; 2Helsinki Institute of Sustainability Science (HELSUS), Faculty of Agriculture and Forestry, University of Helsinki, FI-00100 Helsinki, Finland

**Keywords:** faba bean protein concentrate, *Weissella confusa*, design of experiments, volatile compounds, dextran, multiple linear regression

## Abstract

Fermentation with *Weissella confusa* A16 could improve the flavor of various plant-based sources. However, less is known about the influence of fermentation conditions on the profile of volatile compounds, dextran synthesis and acidity. The present work investigates the synthesis of potential flavor-active volatile compounds, dextran, acetic acid, and lactic acid, as well as the changes in viscosity, pH, and total titratable acidity, during fermentation of faba bean protein concentrate with *W. confusa* A16. A Response Surface Methodology was applied to study the effect of time, temperature, dough yield, and inoculum ratio on the aforementioned responses. Twenty-nine fermentations were carried out using a Central Composite Face design. A total of 39 volatile organic compounds were identified: 2 organic acids, 7 alcohols, 8 aldehydes, 2 alkanes, 12 esters, 3 ketones, 2 aromatic compounds, and 3 terpenes. Long fermentation time and high temperature caused the formation of ethanol and ethyl acetate and the reduction of hexanal, among other compounds linked to the beany flavor. Levels of dextran, acetic acid, and lactic acid increased with increasing temperature, time, and dough yield. Optimal points set for increased dextran and reduced acidity were found at low temperatures and high dough yield. Such conditions would result in hexanal, ethyl acetate and ethanol having a relative peak area of 35.9%, 7.4%, and 4.9%, respectively.

## 1. Introduction

Current environmental and health concerns [1,2] have led to reduced meat consumption and the need for various nutritious plant-based products [3]. Among pulses, the faba bean represents a promising alternative to meat owing to its nutritional, agroecological, and technological advantages [4,5]. Faba bean protein concentrate (FPC) is a product having high protein content (~60%) [6]. It is made by air-classification of the milled seeds, a sustainable process that produces protein-rich pulse ingredients [7]. Because of its high protein content, FPC can be processed into texturized vegetable proteins [6,8,9]. These are products that have a meat-like texture and that have the potential to replace meat. However, the flavor of such extrudable plant-based ingredients is often seen as challenging for consumers’ acceptance and liking [10,11].

The flavor of FPC was described as bitter, astringent, and pea-like [6]. The pea odor and flavor in FPC were linked to the presence of lipid-degrading enzymes (lipase and lipoxygenase) and several lipid-oxidation products, such as 1-hexanol. This and other compounds (e.g., hexanal, octanal, nonanal, 1-octen-3-ol, octanol) are very common in different protein-rich materials [10] and are often associated with beany, earthy, and grass odors [12]. The combination of several volatile compounds is known to result in an overall beany odor [13,14].

Most often, the off-flavors of plant-based meat alternatives are masked by additional ingredients [15], which could hinder consumer demand for clean labels [16,17]. On the other hand, fermentation as a bioprocessing tool is also a standard procedure in the food sector, and could represent a sustainable alternative [18] to mask off-flavors. Indeed, microorganisms can be employed to synthesize several volatile flavor compounds and have great commercial potential in food applications [19]. Lactic acid bacteria have been found to significantly alter the volatile profile of plant protein ingredients, thus achieving sensory improvement [20,21,22]. Among lactic acid bacteria, *Weissella confusa* is one of the most studied for synthesizing functional dextran in various food applications. The dextran produced by *W. confusa* improved the quality of several plant-based products [23,24,25]. It seems to be a viable option for enhancing the sensory appeal of faba bean products. Furthermore, due to its several functional traits, *W. confusa* was recently suggested to be included in the Qualified Presumption of Safety status list, according to the EFSA Biohazard protocols [26]. In addition, the fermentation of pulses with lactic acid bacteria, including fermentation of faba bean with *W. confusa*, has been found to reduce the levels of certain antinutrients [27,28,29,30].

However, fermentation has not always been successful as a flavor-masking tool. Kaleda et al. [31] showed that fermentation increased the overall sensory intensity, including that of unwanted attributes (e.g., bitter, sour, cereal) in plant-based meat alternatives. Fermentation, especially for a long time and at high temperatures, can cause drops in pH, formation of organic acids (e.g., acetic acid and lactic acid), and synthesis of unpleasant volatile organic compounds [31,32]. Even though the high sourness produced could suppress the bitterness, it would still create a barrier to consumers for certain food products, such as meat alternatives. However, in other food applications (e.g., plant-based yogurts), the sourness caused by fermentation could improve the overall sensory quality [33]. In the case of products for which sourness is not a desired quality, optimization of the fermentation process has to be considered when aiming to improve the flavor of plant-based ingredients [10,24,31].

Moreover, alterations to the sensory profile caused by fermentation are dependent on both the substrate and the strain [13]. Therefore, understanding the metabolism of the selected starter culture on a specific food matrix, as affected by fermentation conditions, is also important. Currently, studies on modifications to the volatile profile brought on by *W. confusa* during the fermentation of faba bean ingredients are lacking in the literature.

In this context, the present study aimed at (I) characterizing the volatile profile of faba bean protein concentrate fermented by *W. confusa* A16 at varying fermentation times, temperatures, dough yields, and inoculum ratios; (II) predicting the formation or degradation of volatile compounds as affected by fermentation conditions; (III) investigating the effect of fermentation conditions on the synthesis of dextran, acetic acid, and lactic acid; (IV) finding optimal set points for maximized dextran production and minimized acidity.

## 2. Materials and Methods

### 2.1. Design of Experiment (DoE)

A Response Surface Methodology (RSM) was used to study the effect of fermentation conditions on the synthesis of volatile compounds, dextran, acetic acid, lactic acid, and viscosity, pH, and total titratable acidity (TTA). The studied fermentation parameters were time (10 h, 17 h, 24 h), temperature (20 °C, 25 °C, 30 °C), dough yield (flour to water ratio 200, 300, 400), and starter inoculum ratio (5 log cfu/g, 6 log cfu/g, 7 log cfu/g). Lower and upper limits were defined based on previous experience with this study’s strain and matrix [24,25]. A Central Composite Face (CCF) design was computed using the software MODDE^®^ 13 (Sartorius Corporate Administration, Göttingen, Germany), and 29 experiments were performed, five of which were replicate center points. The experimental design matrix is shown in Table 1.

Fermentations were conducted as described in 2.2. The fermented samples were stored either at −20 °C (for volatile analysis, pH, and TTA measurements) or −70 °C before freeze-drying. Freeze drying was conducted using a Christ Alpha 1-2 freeze-dryer (Martin Christ Gefriertrocknungsanlagen GmbH, Osterrode am Harz, Germany) at <1 mbar. Freeze-dried samples were then analyzed for dextran, acetic acid, and lactic acid contents.

### 2.2. Preparation of Faba Bean Sourdough

Faba bean protein concentrate was purchased from Suomen Viljava OY (Helsinki, Finland). Its chemical composition and volatile profile were previously described [6]. The dextran-producing *W. confusa* A16, available at the Department of Food and Nutrition, University of Helsinki, Finland, was used as a starter [24]. The strain was routinely cultivated in MRS broth (Neogen^®^, Ayr, UK) in microaerophilic conditions at 30 °C for 24 h. The starter cell density in MRS broth was defined with OD measurements at 600 nm and confirmed by plate counts in MRS agar (Neogen^®^, Ayr, UK). The cell density of the 24-h MRS culture corresponded to 10^9^ CFU/mL. For sourdough preparation, an aliquot of the incubated cell culture was centrifuged (10,000× *g* 10 min) and resuspended in an aliquot of the distilled water used for the sourdough preparation before the inoculum, according to the Table 1. Different initial inoculum ratios were targeted based on the experimental design, as explained in 2.1 and Table 1. The mixtures of faba bean concentrate, water, and sucrose were prepared according to Table 1.

### 2.3. Volatile Analysis

Volatile organic compounds were measured in triplicates using headspace solid-phase microextraction gas-chromatography mass-spectrometry (HS-SPME GC-MS), according to a previously described method [6]. Sourdoughs (2 g) were measured into 20-mL amber SPME vials (La-Pha-Pack, Langerwehe, Germany) and put on the HS-SPME (combiPAL, CTC Analytics, Lake Elmo, MN, USA) tray that was set at 4 °C. A divinylbenzene/carboxen/polydimethylsiloxane fiber (1 cm, 50/30 µm phase thickness; Supelco, Sigma Aldrich, St. Louis, MO, USA) was used for extraction. Samples were incubated for 10 min and extracted for 30 min with agitation at 250 rpm. Incubation and extraction temperature were set at 50 °C. A gas chromatograph (HP 6890 series, Agilent Technologies Inc., Santa Clara, CA, USA) equipped with a MS detector (Agilent 5973 Network, Agilent Technologies Inc., Santa Clara, CA, USA) was set up for separation using a mid-polar/polar SPB-624 column (31 m × 0.25 mm × 1.4 µm; Sigma Aldrich, St. Louis, MO, USA), splitless injection, and helium as carrier gas (0.7 mL/min initial flow rate). The following temperature profile was used: 40 °C (5 min hold), 200 °C (5 °C/min), and 200 °C (10 min hold). MS was carried out at a scan range of *m*/*z* 40–300 amu in an electron ionization mode at 70 eV at 230 °C (ion source temperature) and 250 °C (quadrupole temperature). Peaks were manually integrated, and identification was carried out using Wiley’s library results (Wiley 7N, Wiley Registry of Mass Spectral Data, 7th Edition). The Linear Retention Indexes (LRI) were calculated based on the retention times of the alkane series 7–30 (Sigma Aldrich, St. Louis, MO, USA) and hexane (Sigma-Aldrich, Schnelldorf, Germany). Reference LRI are shown in Appendix A. Results were expressed as relative peak area (%) of all integrated peaks for each measurement. Throughout the manuscript, the terms “increase” and “reduction” of volatile compounds are intended to describe the changes in the relative proportion in a measured point.

### 2.4. Viscosity Measurement and Determination of Dextran

Viscosity was measured using a rotational rheometer (Rheolaab QC, Anton Paar GmbH, Graz, Austria) equipped with a ST22.02-4V probe (Rheolaab QC, Anton Paar FmbH, Graz, Austria), as previously described [24]. Measurements were conducted before (at three different dough yields) and after fermentation at a shear rate ranging from 2 s^−1^ to 100 s^−1^. Viscosity was calculated as the ratio of relative viscosity to the unfermented samples, and values were presented as Pa·s.

Dextran levels were determined in freeze-dried samples (100 mg) using a high-performance anion exchange chromatography with a pulse amperometric detection (HPAEC-PAD) system as previously described [34]. The HPAEC-PAD system was equipped with a CarbPac PA-1 analytical column (4 × 250 mm, Dionex Corporation, Sunnyvale, CA, USA), a Waters 2707 autosampler (Waters, Milford, MA, USA), three Waters 515 HPLC pumps (Waters, Milford, MA, USA), and a Waters 2465 pulsed amperometric detector (Waters, Milford, MA, USA). The analyses were performed in gradient mode of Milli-Q water (eluent A; Merck Millipore, Darmstadt, Germany) and 200 mmol/L NaOH (eluent B; Sigma-Aldrich, Schnelldorf, Germany) at a flow rate of 1 mL/min and column temperature of 30 °C. The 300 mmol/L NaOH (eluent C; Sigma-Aldrich, Schnelldorf, Germany) was used for post-column addition (0.3 mL/min). Glucose (Merck KGaA, Darmstadt, Germany) was used as the external standard, and 2-deoxy-D-galactose (Sigma-Aldrich, Schnelldorf, Germany) served as the internal standard for qualification.

### 2.5. Acidity

#### 2.5.1. pH and Total Titratable Acidity Measurements (TTA)

The pH of each sample was determined using a pH meter (Model HI 99161, Hanna Instruments, Woonsocket, RI, USA). TTA was assessed using a pH Titrator (EasyPlus, Mettler Toledo, Columbus, OH, USA) as the amount of 0.1 M NaOH required to get 10 g of sourdough in 100 mL Milli-Q water (Merck Millipore, Darmstadt, Germany) to a pH of 8.5. Measurements were carried out in triplicate according to in-house protocols [24].

#### 2.5.2. Determination of Organic Acids

Acetic and lactic acids were measured in triplicates using high-performance liquid chromatography (HPLC). Briefly, 1 g of freeze-dried sourdough was extracted in 4 mL of 50 mM Tris-HCl buffer (pH 8.8; TRIZA base, Sigma-Aldrich, Schnelldorf, Germany; HCl, Merck Millipore, Darmstadt, Germany) by shaking (1 h, 4 °C) and centrifuging (15 min, 12,000× *g*). Perchloric acid (5%; Fisher Scientific, Loughborough, UK) was added to the extracts in equal volumes (1 mL). Samples were left overnight at 4 °C and then centrifuged again. Extracts were filtered using Amicon^®^ ultra-centrifugal filters (10 K, Merck Millipore, Darmstadt, Germany) and injected into HPLC. Separation and quantification of lactic acid and acetic acid were performed with the method reported by Immonen et al. [35] with small modifications. Briefly, an HPLC system was equipped with a photodiode detector (PDA; Waters 996, Waters Corp., Milford, MA, USA) and a refractive index detector (RI; Waters 2414, Waters Corp., Milford, MA, USA). Elution was carried out using Hi-Plex H column (300 × 6.5 mm; Agilent, CA, USA) at 40 °C with 10 mM H_2_SO_4_ (Sigma Aldrich, St. Louis, MO, USA) as a mobile phase and with a flow rate of 0.5 mL/min. External calibration curves were used for quantitation, and acetic acid was quantified with the RI-detector and lactic acid with PDA.

### 2.6. Model Validity

#### 2.6.1. Model Fit and Internal Validation

MODDE^®^ 13 (Sartorius Corporate Administration, Göttingen, Germany) was used to fit the research data in a Multiple Linear Regression (MLR) model. All factors (fermentation parameters) were orthogonally scaled. Logarithmic transformation was applied to the responses in certain cases to meet the criteria of normal distribution. Model terms were selected based on their significance and contribution to the model quality.

The quality of the model was validated internally by considering the following indicators: R2, Q2, and reproducibility. R2 represents the ability of the model to fit the data, and values closer to 1 indicate a very good model fit. Q2 represents the ability of the model to predict new data, and values closer to 1 indicate very good predictive ability. Reproducibility represents the variation among the replicate center points, and values closer to 1 indicate very good reproducibility.

#### 2.6.2. Optimal Set Points and External Validation

Optimal set points (OSPs) were found using the Optimizer tool available in MODDE^®^ 13 (Sartorius Corporate Administration, Göttingen, Germany). The required criteria for the responses were set as follows: dextran, maximized; acetic acid, minimized; lactic acid, minimized; pH, maximized; TTA, minimized. Viscosity was set as an observed response. Fermentation conditions of three OSPs were predicted, and sourdoughs were prepared as described in 2.2. Samples were then analyzed for dextran content, viscosity, and acidity. The capacity of the model to predict the data was validated according to Cosson et al. [36], and measured responses were compared to the predicted responses. External validation was achieved when measured values fell within the 95% confidence interval of prediction.

### 2.7. Statistical Analysis

Values were expressed as means (± standard deviation) of replicate measurements. MODDE^®^ 13 (Sartorius Corporate Administration, Göttingen, Germany) was used to obtain the following model statistics: degrees of freedom (DF), residual standard deviation (RSD), R2, Q2, reproducibility, and model terms coefficients (showing 95% confidence interval). Principal component analysis (PCA) was carried out using The Unscrambler^®^ X (Version 10.5, Aspen Technology Inc., Bedford, MA, USA). For a balanced dataset, PCA was computed on the relative peak areas of the volatiles detected in samples N1–N24 and the mean of samples N25–29, which were biological replicates of the sourdoughs. For clarity of the interpretation, fermentation parameters were also included in the PCA, but their contribution was downweighed not to influence the model result. PCA was computed using singular value decomposition (SVD) as the algorithm.

## 3. Results and Discussion

### 3.1. Volatile Profiles of Fermented Samples

The volatile profiles of the faba bean sourdoughs are shown in Table 2. The following classes of compounds have been observed: organic acids, alcohols, aldehydes, alkanes, esters, ketones, aromatic compounds, and terpenes. The class with the highest number of compounds identified was esters, followed by aldehydes and alcohols.

The number of volatile compounds drastically increased as a result of fermentation and a total of 39 compounds were detected among the fermented samples. Tuccillo et al. [6] studied the volatile profile of unfermented FPC using the same batch of FPC, storage conditions (−20 °C in the dark), and SPME-GC-MS conditions as in the present study. The authors identified the following volatile compounds in unfermented FPC: 1-hexanol, 2-heptanone, 2-hexenal, 2-pentylfuran, 3-hexen-1-ol, alpha-pinene, delta-3-carene, D-limonene, hexanal, and nonanal. Apart from 2-heptanone, all the aforementioned compounds were also detected in the fermented samples. 2-Heptanone is formed from the oxidation of linoleic acid [37], and its presence was linked to conventional thermally treated faba bean seeds [38] and associated with soapy, beany, and cereal odors [6,39]. In faba bean flour, the levels of 2-heptanone were not affected by storage conditions [40] and actually increased after seed germination [39]. Our study showed that fermentation, even at low a temperature (20 °C) and for a short time (10 h), was successful in the removal of 2-heptanone.

Samples that were fermented for 24 h at 30 °C had increased levels of organic acids, esters, and to a certain extent, the aromatic compounds. However, a reduction in the levels of aldehydes, alkanes, and terpenes was observed. Similarly, samples with the highest dough yield (i.e., 400) had reduced levels of aldehydes and increased levels of esters. However, levels of organic acids decreased, and levels of alcohols and alkanes increased in samples with a dough yield of 400. No clear effect of inoculum ratio on the classes of volatile compounds was observed. As aldehydes contribute to the off-flavors in pulses, their removal/reduction has usually been targeted by enzymatic treatments using aldehyde dehydrogenase [41,42]. Our findings show that fermentation, especially for 24 h at 30 °C, decreased the levels of aldehydes. Degradation of aldehydes was also observed during the fermentation of pea protein isolate by lactic acid bacteria and yeasts [22]. The authors also noticed esters being produced during fermentation and speculated that these compounds could have masked the off-flavors in pea protein isolate.

The score and loading plots of Figure 1 highlight the differences among samples as well as the volatiles and fermentation parameters that contribute to those differences. PC1 explained 75% of the total variance, whereas PC2 explained an additional 21% of the total variance. The majority of samples fermented for shorter time periods (i.e., 10 h) were located on the right side of PC1. In contrast, samples with high fermentation time and high dough yield were located on the left side of PC1 on the lower and upper corners, respectively. The most and least influencing factors were time and inoculum ratio, respectively. Long fermentation time and high temperature were linked with the presence of ethanol, ethyl acetate, 2-methylfuran, ethyl lactate, and hexanoic, heptanoic and octanoic acid ethyl esters. On the other hand, terpenes, aldehydes, alkanes and several alcohols (e.g., 1-butanol-3-methyl, 1-pentanol, 1-penten-3-ol, 1-octanol) and esters (methyl and propyl isovalerate) were linked to shorter fermentation time and lower temperature. Dough yield had a linear relationship with 1-hexanol, 3-hexen-1-ol, isovaleric acid, and acetic acid methyl and hexyl esters.

### 3.2. Effect of Fermentation Conditions on Modelable Volatile Compounds

Among the 39 detected volatiles in faba bean sourdoughs, 14 showed good model validity (R2 > 0.8, Q2 > 0.7); therefore, their levels during fermentation could be predicted. Model statistics and coefficients of the model terms are shown in Table 3. All modellable volatile compounds were modeled as a function of time, temperature, and dough yield. The inoculum ratio only had a significant effect on the levels of 1-penten-3-ol, hexane, and octane. Several square and interaction effects were observed: time × time, dough yield × dough yield, inoculum × inoculum, time × temperature, time × dough yield, temperature × dough yield, and dough yield × inoculum. A linear relationship between time and temperature was observed in all samples.

At increasing fermentation time and temperature, the relative peak area of the following volatiles also increased: ethanol, ethyl acetate, hexanoic acid ethyl ester, ethyl isovalerate, and ethyl lactate. These compounds are typical products of the metabolism of heterofermentative lactic acid bacteria and their effect on the overall flavor of fermented food products has been described [43]. On the other hand, the levels of hexanal, 3-methyl-1-butanol, 2-hexenal, nonanal, 1-pentanol, pentanal, hexane, and octane decreased with increasing fermentation time and temperature. Hexanal, a product of oxidation of linoleic acid, is frequently referred to in the literature as a major contributor to the beany flavor of pulses [13]. Its levels notably decreased during fermentation, thus confirming previous findings that hexanal is reduced during fermentation by lactic acid bacteria [32]. The degradation pathway of hexanal by lactic acid bacteria is still unclear [13]. Our results support the theory that the degradation of hexanal by lactic acid bacteria would yield hexanoic acid ethyl ester as a secondary product [13]. When studying the effect of germination on the volatile profile of faba bean, the levels of hexanal and nonanal decreased until 48 h, but notably increased afterward [44]. According to the authors, the levels of both compounds increased during storage. Another study [14] found that the beany flavor can be enhanced when hexanal is combined with 3-methyl-1-butanol, the levels of which were also reduced during fermentation. During the storage of faba bean flour, the levels of 3-methyl-1-butanol increased [44]. 3-Methyl-1-butanol was described as beany at concentrations of 1–10 ppm, but sweaty and medicinal at 100–10,000 ppm [45]. Because volatile compounds have different odor thresholds and create mutual interactions [46], it is challenging to say whether the removal of a certain volatile compound would result in the total removal of the beany odor. 

Accordingly, it was suggested that more studies are needed to clarify the extent to which hexanal causes the beany odor, while taking into consideration different interaction effects [13]. Such an idea should be applied to several other volatile compounds, including 2-hexenal. A possible relation between this compound and the pea odor and flavor of FPC was previously observed [6]. The compound 2-hexenal is formed in faba beans of Kontu variety from auto- and enzymatic oxidation of α-linolenic acid [37]. However, this compound was not detected in Canadian faba beans [44], indicating that the volatile profile of faba bean is dependent on the genotype [47], among other factors.

A linear relationship between dough yield, time, and temperature was only observed for a few compounds. At increasing dough yield, we could predict increasing levels of ethanol, hexane, octane, ethyl acetate, hexanoic acid ethyl ester, and ethyl lactate. On the other hand, decreasing dough yield was related to increasing levels of 3-methyl-1-butanol, 1-pentanol, 1-penten-3-ol, pentanal, hexanal, hexanal, 2-hexenal, nonanal, and ethyl isovalerate. Lower water activity greatly impacted the synthesis of short-chain fatty acid esters by *Lactococcus lactis* in vitro [48]. So far, little is known about the effect of dough yield on the volatiles produced during fermentation of faba bean. For example, we reported the response plots of a few modellable volatile compounds as a function of dough yield and time (Figure 2).

### 3.3. Effect of Fermentation Conditions on Viscosity and Dextran Production

Dextran content and viscosity as an indicator of dextran production were measured. In the experimental area of this study, viscosity ranged from 0.69 Pa·s to 91.2 Pa·s, whereas dextran content ranged from 0.32 ± 0.03% dm to 5.58 ± 0.25% dm. Dextran levels increased with higher fermentation time, temperature, and dough yield (Figure 3). Inoculum ratio was a significant factor, but it influenced dextran production to a lesser extent compared to other fermentation parameters. A significant negative correlation was observed between dextran production and the following interactions: time × temperature and time × inoculum ratio. 

The production of dextran has been largely documented in *W. confusa* and *Weissella cibaria* [49]. *W. cibaria* 10 M showed the highest dextransucrase expression at 15 °C, whereas temperatures higher than 15 °C were needed for optimal growth [50]. The authors stated that dextran yield was increased without excessive acidification after 2–10 h of incubation, when sourdough was shifted from 30 °C to 6 °C for two days. For dextran production by *W. confusa* C39-2, optimal temperature ranged between 35 to 40 °C, and fast reduction was observed at temperatures higher than 40 °C [51]. In addition, dextransucrase activity is unstable when temperatures higher than 35 °C are reached [52]. Moreover, the pH of the culture medium has a significant impact on its activity [53]. As pH decreases when fermentation temperature increases, the excessive acidification of the sourdough might hinder the production of dextran [34]. Indeed, optimal pH for dextransucrase activity in *W. confusa* C39-2 and *W. confusa* VTT E-90392 was found at 5.4 [51,52].

Higher viscosity was measured when faba bean flour was fermented with *W. confusa VTT* E-143403 compared to *Leuconostoc pseudomesenteroides* DSM 20193, indicating high dextran synthesizing capacity of *W. confusa* in the faba bean matrix [24]. Because the dextran produced by *Weissella* strains has an effect on viscosity, it can be used to alter the rheological properties of fermented food.

### 3.4. Effect of Fermentation Conditions on Acidity

Acidity was assessed by separately considering the levels of acetic and lactic acids and the values of pH and TTA. In the whole experimental set, acetic acid concentration ranged from 0.03 ± 0.01 mg/g to 3.58 ± 0.22 mg/g. Lactic acid was not detected in the experimental samples fermented for 10 h at 20 °C, while the highest values were observed for samples fermented for 24 h at 30 °C (34.3 ± 1.2 mg/g and 33.4 ± 0.2 mg/g). Values for pH ranged from 4.86 ± 0.01 − 0.05 to 6.22 ± 0.03, whereas TTA values ranged from 7.67 ± 0.08 to 23.3 ± 0.02.

For all of the response variables, good model quality was achieved (R2 > 0.91; Q2 > 0.80; Reproducibility > 0.99, 0.94 for pH). The factors (including square effects and interactions) that affected the production of acidity are displayed in Figure 4. The main variable having a positive effect on the production of acidity was time, followed by temperature and dough yield. It is well known in the literature that pH drops with increasing fermentation time and temperature. On the other hand, the effect of dough yield on acidity during fermentation with *W. confusa* is less known. Dough yield was a significant variable in all the acidity-related models, either as such, as a square term or as an interaction with other factors. As shown in the response surface plots (Figure 4), the square effect of dough yield had a negative effect on the production of acetic acid. Because of this, lower acetic acid production could also be achieved with a high dough yield. This is not true in the case of lactic acid, which increased with increasing dough yield. Interestingly, the inoculum ratio did not have a significant effect on acidity. However, for the modeling of acetic acid, the factor inoculum ratio had to be considered for model improvement as its interaction with time was significant.

Acetic and lactic acid are products of the sugar metabolism of lactic acid bacteria, and both play a role in flavor modulation [54]. Acetic acid is produced to a smaller extent compared to lactic acid, and it is often associated with a pungent cider vinegar-like odor, whereas lactic acid has been described as tart and acrid. Their taste thresholds in water differ, being 15 mg/L and 20 mg/L for acetic acid and lactic acid, respectively. Fermentation conditions and strain selection are factors affecting the extent to which organic acids are released [54]. In addition, acetic acid formation by lactic acid bacteria can be adjusted when adding sucrose to the sourdough [55]. When the added sucrose is used by *Weissella* spp. for dextran production, low acetate production occurs, as *Weissella* spp. cannot metabolize fructose [56].

### 3.5. Optimized Fermentation Conditions for External Model Validation

The model was externally validated by performing further fermentations within the design space, measuring the responses, and comparing the measured values with those predicted by the model. As there is a need for optimized fermentation in bioprocessing aimed at masking flavors [24,25,31], three optimal set points were found for maximized dextran levels and minimized acidity. The fermentation conditions used to make optimized sourdoughs are shown in Table 4. Considering the design space of this study, optimized fermentation was achieved with low fermentation time and high dough yield. The temperature and inoculum ratio were close to the center points.

Apart from the three cases in OSP 2 and OSP 3, the measured values were within the 95% confidence interval of prediction (Table 4). Considering this, we were able to externally validate the model. Among the OSPs, OSP 2 had the highest levels of organic acids and the lowest pH, whereas OSP 3 had the lowest levels of dextran and viscosity. OSP 1 appeared to be a compromise between OSP 2 and OSP 3. The sample fermented with the conditions of OSP 1 showed a cell density of viable lactic acid bacteria of ca. 7.0 log CFU/g and 9.2 log CFU/g before and after fermentation, respectively [57]. The authors confirmed the hypothesis that optimized fermentation of faba bean protein concentrate by *W. confusa* A16 can be employed as a flavor-masking strategy. Moreover, the present paper proposed a methodological approach to optimize fermentation conditions to reduce sourness perception in the sensory profile of different fermented products [24,31].

### 3.6. Overall Effect of Fermentation Conditions on Metabolites

When approaching fermentation as a bioprocessing tool for flavor improvement, fermentation conditions must be properly selected. Among other factors (e.g., faba bean variety), we pointed out that time, temperature, and dough yield are influencing factors when it comes to the synthesis of volatile compounds, dextran, and organic acids by *W. confusa* A16. The amount of dextran and the acidity of the sourdoughs seemed to be correlated and affected by the fermentation conditions in similar ways. To obtain 5% of dextran, FPC needs to be fermented for a long time at high temperatures, and the sourdough should have a dough yield higher than 320. Such conditions would result in very acidic sourdough, having pH lower than 5.6 and acetic and lactic acid contents higher than 2.5 mg/g and 12 mg/g, respectively. Excessive acidity would counteract the flavor-improving quality of dextran [24], and for this reason, optimization was needed. At the optimized fermentation conditions (11.4 h, 23 °C, dough yield 380, inoculum ratio 6.8 log CFU/g), 4% of dextran was predicted to be synthesized in a slightly sour sourdough (pH 6, acetic acid 1.8 mg/g, lactic acid 2.9 mg/g). At those optimal conditions, the relative peak area of each modellable volatile compound would be the following: ethanol 4.9%, 3-methyl-1-butanol 0.6%, 1-pentanol 1.2%, 1-penten-3-ol 0.1%, pentanal 0.2%, hexanal 35.9%, 2-hexenal 0.8%, nonanal 0.2%, hexane 0.2%, octane 0.6%, ethyl acetate 7.4%, hexanoic acid ethyl ester 0.1%, ethyl isovalerate 0.1%, ethyl lactate 0.1%.

Optimizing the model for reduced *“beaniness”* was made challenging because not all the detected volatile compounds in this study had good model validity, as the peak area of more susceptible volatiles can vary among replicate measurements, resulting in poor model quality. However, we were able to model several volatile compounds that have been linked to pea and cereal odors and flavors in our previous research [6]. Because of this, we predicted how the relative peak area of certain volatile compounds was affected by fermentation parameters. For instance, at the aforementioned conditions needed to obtain 5% dextran, the relative peak area of 2-hexenal and 3-methyl-1-butanol, which were linked to pea odor and flavor [6], would be lower than 0.5% and 0.3%, respectively. Such findings align with previous research indicating that fermentation of pulses at high temperatures and for longer times has a clear effect on the volatile profile, as it can reduce the levels of compounds responsible for the beany flavor and synthesize other flavor-active compounds. For instance, fermentation of soy protein isolate by *Lactobacillus helveticus* at 37 °C for 24 h and 48 h, showed a significant decrease in beany odor and flavor [58]. The low pH caused by high temperature and long-term fermentation changes the metabolism of lactic acid bacteria, which begin consuming amino acids and converting them to flavor-active volatiles. [55]. 

However, the effect of fermentation on the volatile profile and the overall flavor is not only affected by fermentation parameters but is also strain- and matrix-dependent. A 24 h fermentation of pea protein isolate by *Lactobacilli* was linked with decreased pea odor, but a 48-h fermentation was linked with cheesy, acid, and salty flavors [20]. Contrarily, lupin flour fermented by *Lactobacillus delbrueckii* subs. *bulgaricus* and *Streptococcus thermophilus* for 20 h at 30 °C caused an increase in compounds (e.g., hexanal) responsible for the beany and green odor [59]. To our knowledge, ours is the first study regarding FPC fermented with *W. confusa* A16 that displayed how several volatiles were affected by fermentation conditions. However, further research is still needed to find odor thresholds of volatile compounds responsible for the beany flavor in FPC. Moreover, we believe that total suppression of such compounds is not enough for producing mild-tasting FPC and that the overall effect of fermentation conditions on flavor precursors and flavor-active compounds needs to be considered. For this reason, we want to highlight the underlying factor that flavor in extrudable plant-based ingredients is complex and challenging at the same time [10] and that further research in the field is needed.

## 4. Conclusions

In an ideal scenario that produces faba bean-based meat alternatives, FPC needs to be processed to reduce its bitter taste and beany flavor, and fermentation could represent a possible solution. This research used a Response Surface Methodology approach with a Central Composite Face design to study the effect of fermentation conditions (time, temperature, dough yield, and inoculum ratio) on the synthesis of volatile compounds, dextran, and organic acids by *W. confusa* A16 in faba bean protein concentrate. This study showed that the volatile profile of faba bean protein concentrate was highly influenced by fermentation conditions. High fermentation temperature and longer fermentation time increased the levels of organic acids, esters, and aromatic compounds and decreased the levels of aldehydes, alkanes and terpenes. The levels of several compounds associated with beany flavor (e.g., hexanal, nonanal, 2-hexenal, 1-butanol-3-methyl) decreased at lower dough yield. Compared to inoculum ratio, the variables with the greatest effect on the synthesis of dextran and organic acids were time, temperature, and dough yield. The model was externally validated, and optimal set points were found for maximized dextran levels and reduced acidity. With a short fermentation period and a high dough yield, optimal fermentation was accomplished.

## Figures and Tables

**Figure 1 foods-11-03579-f001:**
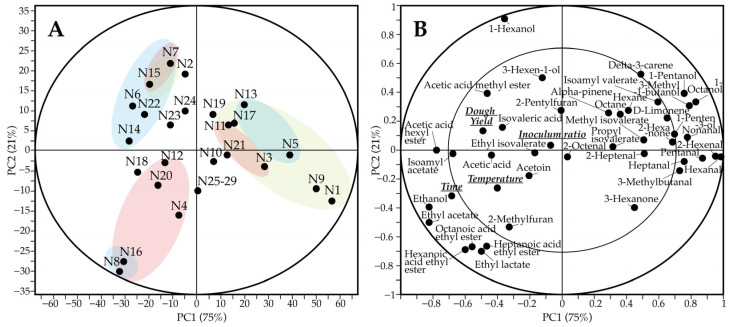
Scores (**A**) and correlation loading plots (**B**) of volatile compounds in fermented faba bean protein concentrate. Green circles—samples fermented for 10 h; red circles—samples fermented at 30 °C; blue circles, samples with dough yield of 400.

**Figure 2 foods-11-03579-f002:**
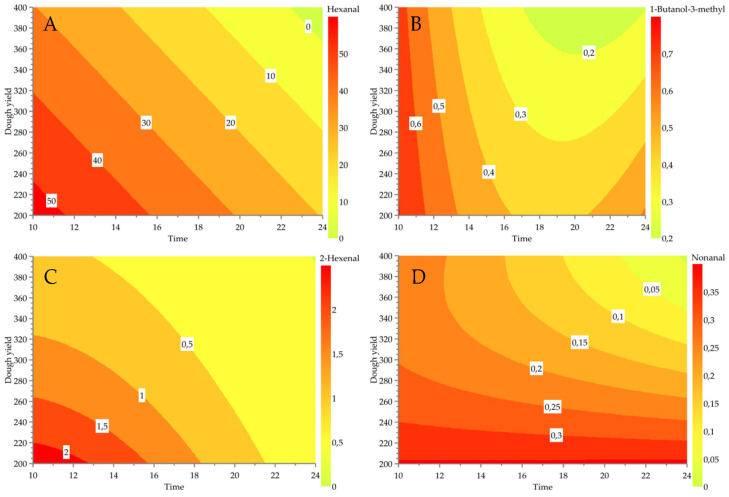
Response surface plots as a function of dough yield and time for (**A**) hexanal, (**B**) 3-methyl-1-butanol, (**C**) 2-hexenal, and (**D**) nonanal. Temperature and inoculum ratio were left constant at 25 °C and 6 log CFU/g, respectively.

**Figure 3 foods-11-03579-f003:**
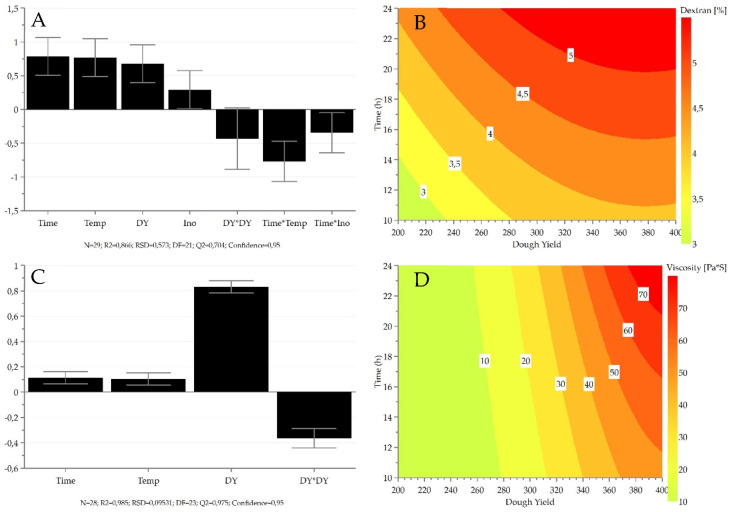
Coefficient plots showing the regression coefficients for the measured responses of dextran (**A**) and viscosity (**C**). Bars indicate the variation in response when a factor changes from 0 to high. The higher the bar, the stronger the effect of the variable on the indicated response. Positive and negative bars indicate a positive and negative effect, respectively. Error bars indicate the 95% confidence interval. (**B**,**D**) show the response surface plots as a function of dough yield and time for dextran and viscosity, respectively. Temperature and inoculum ratio were left constant at 25 °C and 6 log CFU/g, respectively. Temp—temperature; DY—dough yield; Ino—inoculum ratio; N—number of observations; R2—explained variation; RSD—residual standard deviation; DF—degrees of freedom; Q2—predicted variation.

**Figure 4 foods-11-03579-f004:**
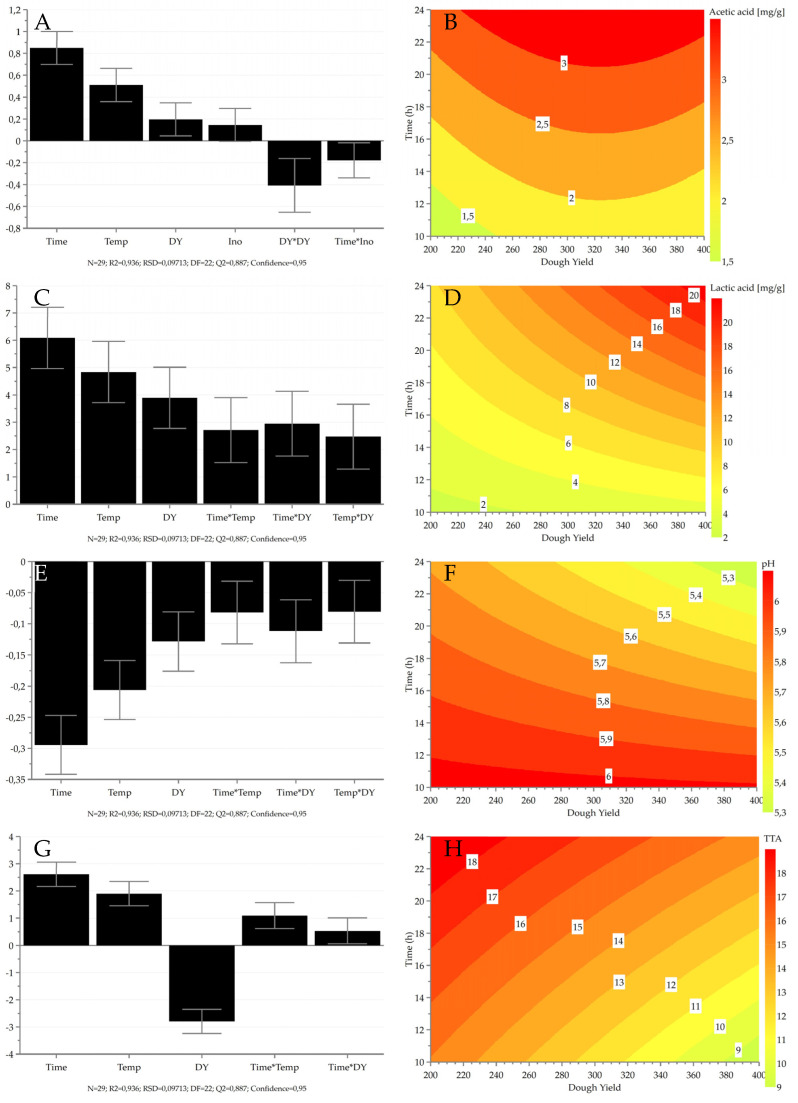
Coefficient plots showing the regression coefficients for the measured responses of acetic acid (**A**), lactic acid (**C**), pH (**E**), and total titratable acidity (**G**). Bars indicate the variation in response when a factor changes from 0 to high. The higher the bar, the stronger the effect of the variable on the indicated response. Positive and negative bars indicate a positive and negative effect, respectively. Error bars indicate the 95% confidence interval. Response surface plots as a function of dough yield and time for acetic acid (**B**), lactic acid (**D**), pH (**F**), and total titratable acidity (**H**). Temperature and inoculum ratio were left constant at 25 °C and 6 log CFU/g, respectively. Temp—temperature; DY—dough yield; Ino—inoculum ratio; N—number of observations; R2—explained variation; RSD—residual standard deviation; DF—degrees of freedom; Q2—predicted variation.

**Table 1 foods-11-03579-t001:** Experimental design matrix: fermentation parameters and sourdoughs recipes.

Experiment Number	Fermentation Parameters				Sourdough Recipes			
	Time (h)	Temperature (°C)	Dough Yield	Inoculum Ratio (CFU/g)	FPC (g)	Water (mL)	Sucrose (g)	*W. confusa* A16 ** (µL)
N1	10	20	200	10^5^	90	100	10	30
N2	24	20	200	10^5^	90	100	10	30
N3	10	30	200	10^5^	90	100	10	30
N4	24	30	200	10^5^	90	100	10	30
N5	10	20	400	10^5^	90	300	10	50
N6	24	20	400	10^5^	90	300	10	50
N7	10	30	400	10^5^	90	300	10	50
N8	24	30	400	10^5^	90	300	10	50
N9	10	20	200	10^7^	90	100	10	2000
N10	24	20	200	10^7^	90	100	10	2000
N11	10	30	200	10^7^	90	100	10	2000
N12	24	30	200	10^7^	90	100	10	2000
N13	10	20	400	10^7^	90	300	10	4000
N14	24	20	400	10^7^	90	300	10	4000
N15	10	30	400	10^7^	90	300	10	4000
N16	24	30	400	10^7^	90	300	10	4000
N17	10	25	300	10^6^	90	200	10	300
N18	24	25	300	10^6^	90	200	10	300
N19	17	20	300	10^6^	90	200	10	300
N20	17	30	300	10^6^	90	200	10	300
N21	17	25	200	10^6^	90	100	10	200
N22	17	25	400	10^6^	90	300	10	400
N23	17	25	300	10^5^	90	200	10	40
N24	17	25	300	10^7^	90	200	10	3000
N25 *	17	25	300	10^6^	90	200	10	300
N26 *	17	25	300	10^6^	90	200	10	300
N27 *	17	25	300	10^6^	90	200	10	300
N28 *	17	25	300	10^6^	90	200	10	300
N29 *	17	25	300	10^6^	90	200	10	300

FPC—faba bean protein concentrate; *—experiment’s replicate center points. **—an aliquot (µL) of 24 h cell culture used to inoculate the sourdough to target the corresponding Inoculum Ratio.

**Table 2 foods-11-03579-t002:** Volatile compounds in faba bean protein concentrate as affected by time, temperature, dough yield, and inoculum ratio during fermentation by *Weissella confusa* A16. Refer to Table 1 for the fermentation parameters of each sample (N1–N24 and the mean of samples N25–29). Number of replicates N = 3.

Volatile Compounds	LRI	Relative Peak Area (%)																								
		N1	N2	N3	N4	N5	N6	N7	N8	N9	N10	N11	N12	N13	N14	N15	N16	N17	N18	N19	N20	N21	N22	N23	N24	N25-29
**Organic Acids**																										
Acetic acid	752	-	12.3	11.8	10.9	-	5.0	4.5	7.4	0.9	8.6	11.9	11.5	4.0	7.4	5.7	8.6	10.3	10.1	7.6	11.9	8.5	5.2	6.9	9.2	7.6
Isovaleric acid	959	-	6.8	5.2	5.4	-	2.6	3.1	3.0	1.5	4.6	4.3	4.7	2.8	3.9	3.5	3.3	4.9	3.7	3.5	3.8	4.1	2.5	3.0	2.9	2.9
**Alcohols**																										
Ethanol	3.56 **	0.7	6.7	2.0	16.7	0.3	11.6	7.3	16.6	1.0	10.5	4.5	19.2	1.3	11.4	8.7	17.6	4.6	13.3	6.7	15.6	7.4	10.5	12.6	8.3	12.6
3-Methyl-1-butanol	813	0.7	0.7	0.7	0.3	0.7	0.4	0.6	0.1	1.0	0.6	0.4	0.5	0.8	0.3	0.5	0.1	0.8	0.1	0.4	0.1	0.5	0.2	0.3	0.2	0.3
1-Pentanol	845	1.7	1.5	1.1	0.7	1.6	0.7	1.3	0.2	1.4	1.0	1.3	0.6	1.8	0.5	1.1	0.1	0.7	0.4	0.8	0.4	1.1	0.5	0.6	0.5	0.4
1-Penten-3-ol	756	0.3	0.3	0.3	0.2	0.1	0.1	0.1	0.01	0.3	0.2	0.2	0.1	0.1	0.03	0.1	0.01	0.1	0.03	0.1	0.04	0.2	0.1	0.1	0.1	0.1
3-Hexen-1-ol	939	-	0.5	-	0.6	-	0.3	0.6	0.1	-	0.5	-	0.2	0.9	0.3	0.4	0.1	0.7	0.1	0.7	0.2	0.6	0.2	0.4	0.5	0.4
1-Hexanol	947	11.7	45.2	19.8	15.8	23.4	46.4	52.0	11.7	14.7	25.1	31.5	28.0	37.4	38.8	49.6	11.9	32.3	29.0	37.1	24.7	26.1	44.1	38.7	40.1	22.5
1-Octanol	816	0.4	0.3	0.4	0.2	0.6	0.3	0.3	0.1	0.6	0.3	0.4	0.4	0.5	0.2	0.3	0.1	0.4	0.1	0.3	0.1	0.4	0.2	0.3	0.3	0.3
**Aldehydes**																										
3-Methylbutanal *	717	0.2	0.1	0.4	0.1	0.4	0.1	0.05	0.02	0.3	0.1	0.1	0.1	0.1	0.1	0.03	0.02	0.1	0.04	0.03	0.1	0.1	0.01	0.05	-	0.03
Pentanal	759	0.6	0.1	0.5	0.2	0.9	0.1	0.2	0.1	0.5	0.3	0.3	0.1	0.4	0.1	0.1	0.05	0.2	0.1	0.1	0.1	0.2	0.1	0.1	0.1	0.1
Hexanal	865	77.3	16.7	48.3	18.3	59.5	1.9	13.9	0.1	70.7	29.8	34.4	11.9	41.5	0.7	7.3	-	37.1	2.2	30.0	10.2	35.2	6.9	15.5	20.1	27.1
2-Hexenal	936	2.6	0.6	1.8	-	1.7	-	0.5	-	2.8	0.6	1.7	0.2	1.0	-	0.2	-	1.1	0.03	0.7	0.2	1.0	0.1	0.6	0.6	0.9
Heptanal	971	0.3	0.1	0.2	0.1	0.3	0.2	0.04	-	0.3	0.2	0.1	-	-	-	0.1	-	0.1	-	-	-	0.1	-	-	-	-
2-Heptenal	1045	0.5	0.3	0.4	0.6	0.1	-	0.2	0.02	0.3	0.5	0.5	0.3	0.3	0.1	0.2	0.04	0.1	0.1	0.3	0.3	0.7	0.1	0.2	0.3	0.4
2-Octenal	1152	-	0.3	-	0.6	-	-	-	-	-	0.6	0.4	0.4	-	0.02	-	-	0.2	0.1	0.3	0.3	0.7	-	-	-	-
Nonanal	1182	0.4	0.4	0.3	0.4	0.4	-	0.2	-	0.4	0.4	0.3	0.2	0.2	-	0.1	-	0.2	0.1	0.3	0.2	0.4	0.1	0.1	0.2	0.2
**Alkanes**																										
Hexane	620	0.1	0.1	0.1	-	0.7	0.1	0.2	-	0.05	0.03	0.1	-	0.3	0.1	0.1	-	0.2	-	-	-	-	-	-	-	-
Octane	823	0.3	0.3	0.3	0.1	2.6	0.4	0.7	0.1	0.2	0.1	0.2	0.1	0.9	0.3	0.6	0.03	0.4	0.03	0.2	0.1	0.1	0.2	0.2	0.2	0.1
**Esters**																										
Ethyl acetate	664	-	1.6	0.6	19.3	-	21.9	9.1	43.4	-	7.8	1.6	13.9	1.4	26.4	16.1	38.1	1.5	23.3	5.6	20.7	6.4	21.9	15.0	11.3	18.6
Isoamyl acetate	931	-	0.1	0.2	0.3	-	0.4	0.1	0.2	-	0.3	0.1	0.2	0.1	0.4	0.2	0.2	0.04	0.3	0.1	0.3	0.2	0.3	0.2	0.2	0.3
Acetic acid methyl ester	4.49 **	0.04	0.1	0.3	0.3	0.05	0.5	0.4	0.3	0.1	0.3	0.3	0.3	0.7	0.5	0.5	0.3	0.3	0.3	0.4	0.3	0.4	0.4	0.4	0.4	0.4
Acetic acid hexyl ester	1070	-	0.6	0.3	1.1	-	2.2	0.8	1.2	-	1.0	0.5	0.8	0.5	2.5	1.3	1.5	0.3	3.4	1.3	1.9	1.0	2.4	1.5	1.5	1.5
Hexanoic acid ethyl ester	1054	-	-	-	2.2	-	1.0	-	6.3	-	0.4	-	1.0	-	1.6	-	5.9	-	5.1	0.9	2.5	-	1.2	0.6	0.3	0.9
Heptanoic acid ethyl ester	1156	-	-	-	0.1	-	-	-	0.2	-	-	-	-	-	-	-	0.3	-	0.2	-	0.1	-	-	-	-	-
Octanoic acid ethyl ester	1257	-	-	-	0.7	-	0.2	-	1.0	-	0.2	-	0.1	-	0.4	-	1.5	-	1.2	0.1	0.5	0.1	0.2	-	-	-
Isoamyl valerate *	1163	0.4	0.6	0.6	0.3	0.4	0.3	0.3	0.1	0.5	0.8	0.6	0.3	0.3	0.3	0.2	0.1	0.5	0.2	0.5	0.2	0.6	0.2	0.4	0.5	0.4
Methyl isovalerate	827	0.1	0.1	0.1	0.1	0.1	0.1	-	0.02	0.1	0.2	0.1	0.1	0.1	0.1	-	-	0.05	0.04	0.04	-	0.2	0.04	0.05	0.03	0.1
Ethyl isovalerate	903	-	0.8	0.1	0.8	-	0.4	0.1	0.2	-	1.8	0.4	0.6	-	0.4	0.2	0.2	0.1	0.3	0.3	0.3	1.2	0.3	0.3	0.2	0.4
Propyl isovalerate	1202	-	0.1	0.4	0.1	0.1	0.03	0.01	-	0.03	0.1	-	-	0.02	0.04	0.04	-	0.04	-	0.1	-	0.05	0.01	-	-	-
Ethyl lactate	889	-	0.03	-	0.8	-	0.4	0.1	5.8	-	0.1	-	0.3	-	0.8	0.2	6.1	-	1.8	0.1	1.1	0.1	0.3	0.1	0.1	0.3
**Ketones**																										
Acetoin	805	-	0.04	0.3	0.5	-	0.03	-	0.2	0.02	0.04	0.1	2.5	-	0.2	0.04	0.1	0.02	0.6	-	0.4	0.02	-	-	0.1	0.03
2-Hexanone *	859	0.1	0.1	0.1	-	0.1	-	-	-	0.1	-	-	-	0.1	-	0.1	-	-	-	-	-	-	-	-	-	-
3-Hexanone *	851	0.1	0.04	0.1	0.05	0.1	-	-	0.1	0.1	-	0.1	-	0.04	0.03	0.04	0.1	-	-	-	-	-	-	-	-	-
**Aromatic Compounds**																										
2-Methylfuran *	655	0.5	0.8	1.0	0.5	1.1	0.3	0.8	2.4	0.5	0.5	1.0	0.3	0.6	0.5	0.5	3.4	1.0	3.1	0.4	1.8	0.3	0.2	0.2	0.3	0.2
2-Pentylfuran	1039	0.4	0.3	0.3	0.6	0.9	0.6	0.6	0.2	0.3	0.5	0.5	0.3	0.5	0.5	0.6	0.4	0.3	0.5	0.4	0.5	0.3	0.5	0.4	0.4	0.4
**Terpenes**																										
Alpha-pinene	975	-	0.1	0.2	0.05	0.5	0.2	0.1	-	-	0.1	0.1	-	-	0.1	0.2	-	0.2	-	-	-	0.1	-	-	-	-
Delta-3-carene *	1056	0.5	1.4	1.9	1.0	2.6	1.3	1.4	-	1.3	1.5	1.9	0.9	1.3	1.1	1.1	-	1.2	-	1.0	1.0	1.2	1.1	1.1	1.2	0.6
D-Limonene	1077	0.4	0.2	0.2	0.2	0.7	0.2	0.2	-	0.2	0.3	0.3	-	0.2	0.2	0.2	-	0.2	0.1	0.5	0.1	0.2	0.1	0.1	0.2	-

LRI—Linear Retention Index; *—coelution with other compounds might have occurred; **—retention time (LRI not determined because the compounds eluted before the retention time of hexane (retention time 4.8 min); -—not determined.

**Table 3 foods-11-03579-t003:** Model statistics and terms’ coefficients of the modellable volatile compounds (R2 > 0.8, Q2 > 0.7). Number of observations *n* = 29.

Modellable Volatile Compounds	Model Statistics					Model Terms Coefficients										
	DF	RSD	R2	Q2	Repr	Tim	Temp	DY	Ino	Tim × Tim	DY × DY	Ino × Ino	Tim × Temp	Tim × DY	Temp × DY	DY × Ino
Ethanol	24	1.85	0.90	0.85	0.98	1.0	0.6	0.2		−0.5						
3-Methyl-1-butanol	23	0.11	0.82	0.72	0.76	−0.7	−0.5	−0.4		0.9				−0.3		
1-Pentanol	23	0.16	0.91	0.85	0.98	−0.7	−0.5	−0.3			1.1			−0.3		
1-Penten-3-ol	19	0.02	0.97	0.92	0.99	−0.4	−0.2	−0.9	−0.2	0.3	0.8		−0.2	−0.1		0.1
Pentanal	24	0.13	0.86	0.77	0.98	−0.9	−0.4	−0.5			0.7					
Hexanal	24	6.19	0.92	0.87	0.98	−0.8	−0.5	−0.6					0.4			
2-Hexenal	24	0.18	0.88	0.82	0.96	−0.9	−0.4	−0.6		−0.4						
Nonanal	23	0.05	0.87	0.77	0.96	−0.5	−0.3	−0.9			0.4			−0.5		
Hexane	21	0.09	0.89	0.74	1.00	−0.7	−0.3	0.3	−0.2	1.2				−0.3	−0.2	
Octane	20	0.15	0.90	0.77	0.99	−0.8	−0.5	0.4	−0.2			0.8	−0.2	−0.3	−0.3	
Ethyl acetate	22	3.50	0.93	0.87	0.99	0.8	0.5	0.6					0.2	0.3	0.2	
Hexanoic acid ethyl ester	22	0.22	0.90	0.84	0.99	0.9	0.3	0.3			−0.6		0.3	0.3		
Ethyl isovalerate	21	0.11	0.91	0.80	0.94	0.9	0.2	−0.4		−0.9	0.5		−0.5	−0.3		
Ethyl lactate	22	0.15	0.93	0.88	0.98	0.8	0.5	0.5					0.3	0.3	0.2	

DF—degrees of freedom; RSD—residual standard deviation—R2—explained variation; Q2—predicted variation; Repr—reproducibility; Tim—time; Temp—temperature; DY—dough yield; Ino—inoculum ratio.

**Table 4 foods-11-03579-t004:** Fermentation conditions of the optimal set points (OSP) for maximized dextran and minimized acidity and their predicted and measured responses (*in italics*).

Fermentation Conditions	OSP 1	OSP 2	OSP 3
Time (h)	11.4	15.1	10.3
Temperature (°C)	23	23	25
Dough Yield	380	400	350.5
Inoculum ration (log CFU/g)	6.8	6.7	6
**Predicted and Measured Responses**			
Dextran (%)	4.0; *4.7*	4.3; *4.9*	3.9; *4.5*
Viscosity (Pa·s)	41.5; *47.7*	51.7; *71.2*	34.3; *20.4 **
Acetic acid (mg/g)	1.8; *2.5*	2.1; *2.7*	1.7; *2.1*
Lactic acid (mg/g)	2.9; *5.7*	7.2; *8.8*	3.1; *4.2*
pH	6.0; *5.8*	5.8; *5.5 **	6.0; *5.9*
TTA	9.1; *10.2*	9.9; *12.6 **	10.0; *10.2*

* Value does not fit within the 95% confidence interval.

## Data Availability

The data presented in this study are available on request from the corresponding author. The data are not publicly available due to confidentiality.

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
