# Peer review of "Fermentation Conditions Affect the Synthesis of Volatile Compounds, Dextran, and Organic Acids by Weissella confusa A16 in Faba Bean Protein Concentrate"

_foods, 2022, doi:10.3390/foods11223579_

Round 1

Reviewer 1 Report

 Tuccillo et al., reported the effects of time, temperature, dough, and inoculum ratio on the abundance VOCs, dextran, and organic acids of faba bean protein concentrate Weissella confusa A16. This an interesting study and the authors have done well using SRM in this study. The finding is of interest to the readership of FOODs. The authors should check the documents for grammar. Also below are some points to address

Line 32 reference is needed after products

Why didn’t the authors use internal standard?

Why NaCl not added to the samples?

Please provide the manufacturer, city and country of all equipment used

The Table 2 is not appropriate to report % area. Rather report the area. %area doesn’t actually depict the abundance of the VOCs. Also, why reporting the RT? You should use alkane series and calculate the Kovats retention index and compare it with literature (NIST).

Coelution with other compounds might have occurred? How????

Reviewer 2 Report

Dear Editor and authors, 

A-Major comment

1-Many strains of W. confusa are opportunistic bacteria. Why did you use it as a starter?

2-Why was this bacterial isolate used in soy production? Why not use a starter mixture to increase flavor and other metabolites?

3-Why was the number of viable bacteria in the final product not calculated? Because the number of bacteria has a significant effect on the quantities of compounds produced.

B-Minor comment 

1-The introduction needs to add some references in order to increase the proportion of modernity in the references. I suggest adding 1-(Verma, D. K., Al-Sahlany, S. T. G., Niamah, A. K., Thakur, M., Shah, N., Singh, S., ... & Aguilar, C. N. (2022). Recent trends in microbial flavour Compounds: A review on Chemistry, synthesis mechanism and their application in food. Saudi Journal of Biological Sciences, 29(3), 1565.‏) , 2-Adesulu-Dahunsi, A. T., Dahunsi, S. O., & Olayanju, A. (2020). Synergistic microbial interactions between lactic acid bacteria and yeasts during production of Nigerian indigenous fermented foods and beverages. Food Control, 110, 106963.

2-Page 3 line 7, How were these conditions obtained? How much gas are there? The authors must provide details.

3-Page 3 line 110, How many viable bacterial cells are in the volumes of the added  inoculum?

4-Some abbreviations are added in the manuscript without mentioning them in advance, such as TTA see line 154, page 4.

5-Viscosity was measured using a rotational rheometer, this method needs to add reference (Yunoki, S., Sugimoto, K., Ohyabu, Y., Ida, H., & Hiraoka, Y. (2019). Accurate and precise viscosity measurements of gelatin solutions using a rotational rheometer. Food Science and Technology Research25(2), 217-226.

5- pH and Total Titratable Acidity Measurements, this method needs to add reference (Al-Sahlany, S. T. G., Khassaf, W. H., Niamah, A. K., & Al-Manhel, A. J. (2022). Date juice addition to bio-yogurt: The effects on physicochemical and microbiological properties during storage, as well as blood parameters in vivo .‏https://www.sciencedirect.com/science/article/pii/S1658077X22000716.

6-Page 6, Figure 4 The y-axis label does not exist.

Round 2

Reviewer 1 Report

The authors have improved the manuscript. However, Table 2 needs to fixed. The authors provided the calculated LRT without including literature LRT (NIST) of the corresponding compounds. The supplementary TABLE 1 is irrelevant and should be deleted. The authors should rather include × 10^xxx to account the absolute numbers reported in the supplementary Table 1. 

Author Response

We would like to thank Reviewer 1 for helping us improving the manuscript and for the useful final advices. 
As suggested by the reviewer, we deleted the original Supplementary Table 1, which was replaced by a table showing the comparison between measured and reference LRI(s) (line 673). The method section and the caption of Table 2 were edited accordingly (line 142). Most of the times, reference LRI values are not reported, especially when the GC temperature program is not linear, as it is true in our study. Moreover, the column that we used (SPB-624) is not very common nor standardized, and the exact NIST reference values for that column were not found. Therefore, we included as reference the LRI values from a paper using the same column (Paradiso, V. M.; Summo, C.; Pasqualone, A.; Caponio, F. Evaluation of different natural antioxidants as affecting volatile lipid oxidation products related to off-flavours in corn flakes. Food Chemistry 2009, 113(2), 543-549.) and the values for a comparable column (OV17) found at flavornet.org. We remind the reviewer that identification of the compounds was performed based on the MS spectra, which was compared to Wiley’s library results (line 139). 
Finally, we further checked and edited the manuscript for fine/minor spell checks. Thank you very much. 

Reviewer 2 Report

Dear Editors,

The authors made all the necessary changes to improve the manuscript, and now I recommend it for publication in its current form.

Author Response

We would like to thank Reviewer 2 for approving the manuscript and for helping us to significantly improve its quality.

The last version submitted was further checked and edited for language, according to the suggestion form Reviewer 2. Moreover, we replace the Supplementary Table 1 according to the suggestion from Reviewer 1. No further changes were made. 

Thank you very much.